# Benchmark Probing: Investigating Data Leakage in Large Language Models

## Abstract

Large language models have consistently demonstrated exceptional performance across a wide range of natural language processing tasks. However, concerns have been raised about whether LLMs rely on benchmark data during their training phase, potentially leading to inflated scores on these benchmarks. This phenomenon, known as data contamination, presents a significant challenge within the context of LLMs. In this paper, we present a novel investigation protocol named **T**estset **S**lot Guessing (**TS-Guessing**) on knowledge-required benchmark MMLU and TruthfulQA, designed to estimate the contamination of emerging commercial LLMs. We divide this protocol into two subtasks: (i) *Question-based* setting: guessing the missing portions for long and complex questions in the testset (ii) *Question-Multichoice* setting: guessing the missing option given both complicated questions and options. We find that commercial LLMs could surprisingly fill in the absent data and demonstrate a remarkable increase given additional metadata (from 22.28% to 42.19% for Claude-instant-1 and from 17.53% to 29.49% for GPT-4).

## 1   Introduction

Large language models (LLMs) have demonstrated exceptional performance across a wide range of NLP tasks, and the NLP community has witnessed the emergence of several impressive LLMs. Notably, there are robust commercial LLMs, including the GPT-* [3, 16] by OpenAI, Claude [1] by Anthropic, and Bard [6] by Google, among others. In addition to these commercial models, there are numerous open-source LLMs, such as Llama [22, 23], MPT [13], Falcon [15], and Chinchilla [8]. However, concerns have arisen regarding these LLMs, primarily related to their extensive training on web data, often at a terabyte scale. This extensive training data may, in turn, potentially overlap with the current benchmark [3, 4, 22, 23], which is also frequently constructed from Internet sources. Research has revealed that pretraining on the testset can artificially inflate performance metrics [18]. Consequently, it becomes imperative for the community to address the detection of potential data contamination in these models.

Despite the pressing need for research on data contamination, there appears to be a scarcity of relevant studies. For current Large Language Models, n-gram based methods are introduced [3, 24, 23] to detect data contamination. To summarize, our approach involves employing n-gram tokenization to partition large documents into smaller shards and assessing their similarity with benchmark data [4, 22]. However, this method is contingent upon having complete access to the training corpus, making it challenging to estimate data contamination for models [3, 16, 6, 1, 10] that do not disclose their training data. Therefore, there is a clear necessity to develop a more robust method for detecting potential contamination in both *open-sourced* and *closed-sourced* Language Models.

In this paper, we introduce a novel investigation protocol referred to as TS-Guessing in two distinct settings: (1) Question-based guessing and (2) Question-multichoice guessing shown in Figure 1. In

Submitted to 37th Conference on Neural Information Processing Systems (NeurIPS 2023). Do not distribute.

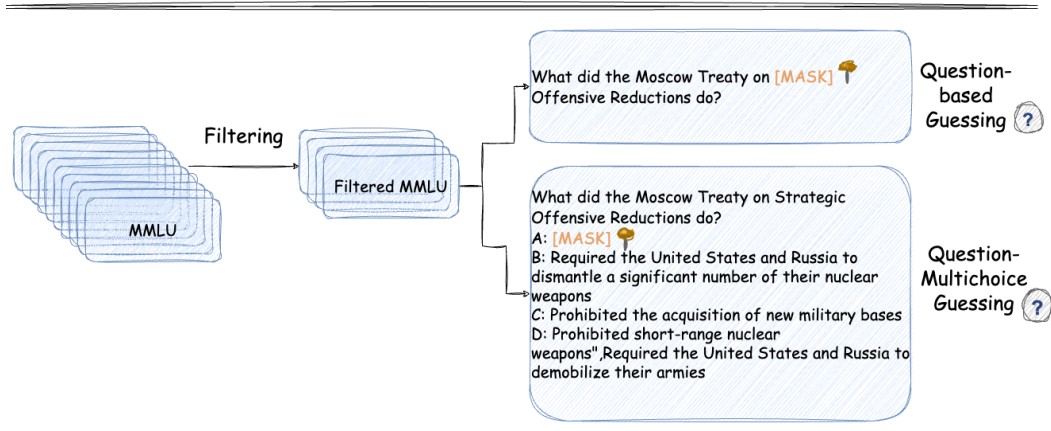

Figure 1: Illasutration of workflow of **TS-Guessing** on MMLU. The prefiltering technique (§ 3.3) filters out correlated and correct options in the benchmark to rationalize our investigation protocol.

the *Question-based* setting, our objective is to hide a crucial word within a sentence, challenging the model to predict it accurately, while avoiding common alternatives from a vast vocabulary. In the *Question-Multichoice* setting, our goal is to conceal an incorrect answer option among multiple choices, preventing the model from giving the correct answer directly and encouraging it to complete the missing part in the benchmark. These two settings guide LLMs in guessing the missing information in the questions and answers, thereby testing their contaminated knowledge against the testset data.

Our experimental results reveal that different versions of LLMs from the *same* company did not exhibit a pronounced difference in their TS-Guessing performance, with GPT-4 showing only a 1% improvement in the zero-shot setting compared to GPT-3.5-turbo, and Claude-2 performing 5% worse than Claude-instant-1 in the question-based guessing task. These findings highlight the consistency in performance among LLMs from the same company and underscore the potential data source similarity. Besides, we observed that commercial large language models (LLMs) achieved a remarkable zero-shot accuracy of 16% with GPT-3.5-turbo, 22% with Claude-instant-1, and 25% with GPT-3.5-turbo in the Question-based setting within TruthfulQA. In the Question-Multichoice setting, GPT-3.5-turbo exhibited a noteworthy ability to guess the missing option, achieving a 57% EM rate. Considering these results, we raise concerns about the potential contamination of the current benchmark, especially if it is made publicly accessible. We urge for this to be seen as a call to action and to explore additional methods to mitigate the risk of contamination.

## 2 Related Work

Recent advancements in LLMs have raised concerns about data contamination and its impact on model performance. To address these concerns, researchers have explored various tokenization strategies and detection methods in the field of natural language processing. Previous research related to LLMs is introduced in GPT-3's Appendix C [3]. In this study, GPT-3 employs 13-gram tokenization for both training data and benchmark data. PaLM [4] also employs an 8-gram strategy, considering data to be contaminated if there is a 70% overlap with 8-grams from the test set data. Open-source models such as Llama [22] adopt a similar methodology, derived from GPT-3. Llama 2 [23] (Section A.6), however, enhances this approach by using 8-gram tokenization with weight balancing. Currently, various methods, including prompt-based and probing-based approaches [5, 11], have been developed to detect potential data contamination in LLMs. Additionally, there are suggestions for mitigating potential leakage when manipulating benchmark test sets [9]. Besides the research conducted on English-only corpora, there is also ongoing investigation [2] into the issue of language contamination in cross-lingual settings.

# 3 Testset Slot Guessing Protocol

## 3.1 What does Question-based and Question-Multichoice stand for?

As shown in Figure 2a, in the context of the *Question-based* setting, our objective is to **mask a pivotal word that represents the essence of the sentence.** Taking the example sentence, "Where did fortune cookies originate?" into consideration, in this instance, the word "fortune" emerges as a potential keyword candidate. This is because predicting the masked sentence, "Where did [MASK] cookies originate?" necessitates the model to select a word from a vast vocabulary list, encompassing options such as "Chocolate chip," "Biscotti," "Snickerdoodle" and so forth. However, if the model has encountered testset data during its training phase, it may exhibit a greater inclination to produce the missing word as 'fortune' rather than "Biscotti." or "Snickerdoodle".

A more challenging task is *Question-Multichoice* setting (shown in Figure 2b). In this particular scenario, **our objective is to mask an incorrect option**. We intentionally *avoid masking the correct option* to prevent the model from directly providing the correct answer, instead compelling it to guess an incorrect answer from a vast set of erroneous possibilities. Furthermore, we implement detailed filtering procedures (introduced in 3.3) to eliminate instances where there exists a strong correlation between any answer options, thereby discouraging the model from relying on its reasoning and inference capabilities to predict the masked words. when confronted with intricate questions and unrelated options, if the model can still output missing options (sometimes exceeding a length of 8) correctly, it raises a compelling suspicion regarding the extent to which the model's behavior is influenced by its exposure to benchmark data.

## 3.2 Problem Formulation

**Question-based**  Let $\mathcal{D}$ be a dataset containing $n$ documents. For each document $d_i$, where $i \in \{1, \ldots, n\}$, there exists a question $q_i$ and several answers. Given a question $q_i$ from document $d_i$, we perform a *keyword searching function*

$$k_i = f_{keyword}(q_i)$$

where $k_i$ is the keyword associated with $q_i$. Subsequently, we use a mask function $q_i' = g(q_i, k_i)$ to mask the keyword in the question with [MASK]. Thus, the overall process can be represented as:

$$q_i' = g(q_i, k_i, [\text{MASK}])$$

**Question-Multichoice**  Let $\mathcal{D}$ be a dataset containing $n$ documents. For each document $d_i$, where $i \in \{1, \ldots, n\}$, there is: A question denoted by $Q$. A list of answers denoted by $A$, where $A = \{a_1, a_2, \ldots, a_m\}$ and $m$ is the number of answers for that document. One correct answer denoted by $a_c$ such that $a_c \in A$.

From the list $A$, one wrong answer is chosen and replaced with [MASK], denoted by $a_{\text{mask}}$. The final template is a concatenation of the question, the correct answer, and three wrong answers (including the masked one):

$$T_i = \text{Concat}\left(Q_i, a_{c_i}, a_{w1_i}, a_{w2_i}, a_{\text{mask}_i}\right)$$

Where $T_i$ is the template for the $i^{th}$ document, $Q_i$ is the question for the $i^{th}$ document, $a_{c_i}$ is the correct answer for the $i^{th}$ document, $a_{w1_i}$ and $a_{w2_i}$ are two wrong answers chosen from the list $A$ for the $i^{th}$ document, $a_{\text{mask}_i}$ is the wrong answer that has been replaced with [MASK] for the $i^{th}$ document.

## 3.3 Experiments Details

**Domains**  We consider two datasets widely recognized for their effectiveness in evaluating knowledge Question Answering in current LLMs benchmarks: (i) **MMLU** [7], a dataset measuring the knowledge capabilities of LLMs and encompasses 57 diverse tasks spanning elementary mathematics, U.S. history, computer science, law, and more. (ii) **TruthfulQA** [14], a benchmark assesses the truthfulness of language models in generating responses to questions across 38 different categories, including health, law, finance, and politics.

**Pre-filtering**  A critical step in our experiment involves the application of filtering techniques. We employ several methods to ensure that our investigative protocol does not become a straightforward

(a) Prompt template of **Question-based** guessing from handpicked example in TruthfulQA.

(b) Prompt template of **Question-Multichoice** guessing from handpicked example in MMLU.

Figure 2: Illustration of two tasks within TS-Guessing. Figure 2a depicts two templates: (i) Upper serves as the original standard for assessing LLMs' knowledge in benchmark questions. (ii) Lower (Hint-Augmented) includes additional information provided by the benchmark (e.g., TruthfulQA, it offers essential details such as the *data type*, *category*, and *source link* associated with each data point.)

semantic inference or logical reasoning task. For TruthfulQA, we implement two filtering criteria: (i) removing data if its question has a length of four words or fewer, and (ii) excluding data associated with the 'Indexical Error' type. For the MMLU dataset, we adopt a more stringent filtering rule, which includes: (i) removing data containing only "Yes-No" or "True-False" options, mathematical symbols, or other simple option expressions; and (ii) removing data if the Rouge-L [12] F1 score between any two options exceeds a predefined threshold. In this paper, we have established a threshold of 0.65 chosed to filter out "three words differing one in a sentence"(e,g, A:"I am American" and B:"I am Swedish." would result in the data being filtered)

**Keyword Searching**   We are implementing a keyword searching function using two powerful tools: the Stanford POS Tagger [21] and ChatGPT with 5-shot in-context learning. Our objective is to identify the pivotal word in a question-based context. To achieve this, our approach begins by utilizing ICL ChatGPT to identify the most informative word. Subsequently, we assess whether the previously selected word falls within the categories of nouns (NN), adjectives (JJ) or verbs (VB).

**Hint**   Hint is employed in the Question-based setting to leverage the supplementary information within the test dataset. In this paper, TruthfulQA not only supplies questions and answer options but also includes additional metadata, such as type, category, and URL information. This metadata serves as an added prompt presented to LLMs. For MMLU, we do not use a hint-based approach since the benchmark consists solely of questions and answers. Nevertheless, we posit that this methodology holds promise for application to other datasets, facilitating the exploitation of information within the test dataset.

### 3.4   Obervations and Analysis

#### 3.4.1   Strong Model Doesn't Indicate Proficiency In TS-Guessing

As depicted in Table 1 and Table 2, despite the increased power of GPT-4, we do not observe significant improvements in our TS-Guessing protocol. In the original version (without hints appended

Table 1: Exact Match (EM) rate in the **Question-based** guessing in TruthfulQA. Three kinds of hints are metadata given in TruthfulQA. (Details in 3.3)

| Model | Company | Question-based | | | |
| --- | --- | --- | --- | --- | --- |
| | | w/o hint | w. type-hint | w. category-hint | w. url-hint |
| GPT-4 | OpenAI | 0.17 | 0.19 | 0.15 | 0.29 |
| GPT-3.5-turbo | OpenAI | 0.16 | 0.17 | 0.19 | 0.25 |
| Claude-2 | Anthropic | 0.23 | 0.25 | 0.25 | **0.37** |
| Claude-instant-1 | Anthropic | 0.22 | 0.23 | 0.21 | **0.42** |

to the prompt), there is only a 1% difference between the two models. Even when utilizing URL-hint prompting in a Question-based setting, the performance gap remains minimal, with only a 4% difference between GPT-3.5-turbo and GPT-4, and a fluctuation of approximately ± 3% in performance in the Question-Multichoice setting. This pattern is consistent in both Claude-instant-1 and Claude-2. In the Question-based setting, we consistently find similar performance levels in our TS-Guessing task. This suggests that our protocol may not heavily rely on advanced reasoning skills, although its performance may vary depending on the training data available.

This phenomenon could be explained in several ways. Firstly, the variance in training data between different companies may be significant. Secondly, even within the same company, different model versions may have closely related training data, especially when considering data that potentially overlaps with the benchmark.

### 3.4.2   Latest Benchmark Could Still Be Comtaminated

As shown in Table 1, there are **16.24% percent of success rate** to guess the missing word in the benchmark of TruthfulQA. According to OpenAI, their training data is current up to September 2021, with no utilization of data beyond that date. However, TruthfulQA made its camera-ready version available on the ACL Anthology in May 2022. Upon closer look, it becomes evident that a substantial portion of the data in TruthfulQA originates from or is derived with assistance from publicly accessible sources. It's worth noting that this publicly available content, particularly when restricted to Wikipedia, remains accessible to commercial AI companies at any given time. Consequently, careful consideration is warranted when assessing potential contamination in new benchmarks.

Table 2: Performance in the **Question-Multichoice** guessing in TruthfulQA. BLEURT is a pre-trained score metrics used in text generation evaluation [19]

| Model | TruthfulQA | | | MMLU | | |
| --- | --- | --- | --- | --- | --- | --- |
| | EM | Rouge-L F1 | BLEURT | EM | Rouge-L F1 | BLEURT |
| GPT-4 | 0.12 | 0.46 | 0.32 | **0.52** | 0.69 | 0.41 |
| GPT-3.5-turbo | 0.10 | 0.43 | 0.30 | **0.57** | 0.67 | 0.44 |

### 3.4.3   MMLU Are Probably Contaminated Seriously

As shown in Table 2, given the fact that we have filtered out the correlated options, mathematical symbol and logic expressions. **GPT-3.5-turbo still could precisely predict masked choices in MMLU testset with 57% accuracy**. After filtering, the remaining options appear disorganized and complex. However, successful examples are rather surprising. In comparison to TruthfulQA, which boasts a 0.10 EM rate and a 0.43 Rouge-L F1 score, the EM rate of MMLU is noticeably higher. The high accuracy suggests that when given a question and the correct answer in MMLU, GPT-3.5-turbo has a probability greater than fifty percent of generating a candidate list with incorrect answers, just like the benchmark. we here could take a successful example in Question-Multichoice Guessing, "Which is not a nonstate actor that poses a threat to the United States?" and a correct answer "D. China" as an example. ChatGPT could magically complete another wrong option "C. Drug traffickers" if we mask option C. The candidate list for a wrong option is large and may even be

infinite, so when seeing LLMs could complete it exactly correctly sometimes for a very long and complex sentence, this raises our concerns of benchmark data leakage.

### 3.5 Corrleation between TS-Guessing and Task Accuracy

As illustrated in Table 3, we have included the *Spearman correlation* as a metric to assess the relationship between our TS-Guessing protocol and task performance, thereby examining the interconnection between these two tasks. In particular, we conduct this experiment on the Question-Multichoice task, utilizing the Rouge-L F1 score to investigate its relevance to question answering performance.

Our findings reveal interesting insights. In the case of TruthfulQA, we observe a negative correlation ($-0.158$ for GPT-4 and $-0.128$ for GPT-3.5-turbo) between task performance and the TS-Guessing protocol. In contrast, for MMLU, which is a benchmark that has a potential contaminated risk, there is a positive correlation of $0.279$ for GPT-4.

Table 3: Spearman correlations between task performance and Rouge-L F1 score. $p < 0.05$ is set default

| Task | Model | Corr. ($\rho$) with... f1 score ↑ |
|------|-------|------|
| TruthfulQA | GPT-4 | -0.158 |
| | GPT-3.5-turbo | -0.128 |
| MMLU | GPT-4 | 0.279 |
| | GPT-3.5-turbo | 0.234 |

We aim to provide an explanation from two perspectives. Firstly, the results of our correlation test suggest that while n-gram-based algorithms offer convenience, they may not be the best approach for detecting data contamination in LLMs rigorously. However, this method is widely used in models such as GPT-3, Llama, and Llama 2 (as discussed in Section 2).

Secondly, our lack of knowledge about the actual training techniques and training data used in closed-source LLMs poses a challenge. In today's landscape, numerous training techniques have emerged, ranging from supervised fine-tuning (SFT) to reinforcement learning from human feedback (RLHF) [17], and even mixture of experts (MoE) [20]. Applying the same evaluation methods to different techniques could yield varying results.

## 4 Conclusion and Future Work

In this paper, we present a novel investigation protocol designed to assess the potential data leakage in benchmark datasets when evaluated with Language Model Models (LLMs). Our results reveal that both commercial LLMs from OpenAI and Claude exhibit the capability to accurately complete missing options in the test set. GPT-3.5-turbo achieved a 57% accuracy in predicting masked choices in the MMLU testset, with remaining options seeming disorganized and complex after filtering. Compared to TruthfulQA, MMLU exhibited a significantly higher EM rate despite its challenging nature. This observation raises concerns about potential data leakage in contemporary benchmark datasets.

This study can be extended beyond closed-source models, encompassing open-source LLMs as well. It offers a valuable tool for identifying and detecting potential data leakage, shedding light on how LLMs acquire knowledge about test data from benchmarks. This, in turn, provides insights into benchmark contamination and informs us about the appropriate times to update our benchmarks. As the field of natural language processing continues to evolve, the development of new benchmark datasets and evaluation protocols should be a priority. These new benchmarks should be designed with robust mechanisms to detect and mitigate data leakage, ensuring the integrity of the evaluation process for future LLMs. Collaborative efforts between researchers, dataset creators, and LLM developers can play a pivotal role in achieving this goal.

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
