# OpenReview forum: "Benchmark Probing: Investigating Data Leakage in Large Language Models"
_NeurIPS.cc/2023/Workshop/BUGS — NeurIPS 2023 BUGS Poster_

### Official Review · Reviewer_bsLG · 2023-10-25
**Is it possible to detect data contamination in LLMs?**

**Rating:** 5
**Confidence:** 4

**Review:**

## Originality and significance
This paper studies whether it is possible to detect data contamination in LLMs. Given that many LLMs do not disclose their training data, it is hard to know if train data is contaminated with test data. Authors propose having a language model complete questions/answers from the test set to evaluate whether they were trained on test data.

*Relevance to workshop*: This submission *does not* appear relevant to the workshop purpose of backdoor research.

## Clarity
- “EM rate” is not defined in the text. It is defined until Table 1.
- Section 3.5 (correlation between TS-guessing and task accuracy) lacks detail. We only have one “task accuracy” for each model, so I am not sure what is being measured here.

## Quality
- In the Question-based setting, I am not convinced that masking a “pivotal word” is related to whether the model has seen the sentence. While the authors have a keyword searching procedure to come up with these “pivotal words,” some words are simply more likely than other words, and I am not convinced that this Question-based setting evaluates whether the model was trained on these questions or not.
- On the other hand, Question-multichoice is interesting because the model must complete an additional incorrect option. There is no incentive for generating reasonable text. I see this setting as a more proper evaluation.

## Summary
Positives:
- Question-multichoice setting is a novel way of evaluating data contamination, and many other variants can be created for new research
- Interestingly, GPT-3.5-turbo and GPT-4 are shown to have more than 52% exact match with incorrect options of multiple choice test set questions.

Negatives:
- Topic of paper does not seem related to workshop
- Question-based setting evaluation is not convincing. More explanation is needed as to why a correct completion shows the model has seen the question before. Unlike Question-multichoice, there is *not* an infinite number of good choices for a word.
- Section 3.5 is not understandable. If the authors are assessing the relationship between TS-guessing and task performance, they would need different models with different TS-guessing scores and associated task evaluations, but there are only 2 models per benchmark.

---

### Official Review · Reviewer_6v7S · 2023-10-27
**Interesting direction**

**Rating:** 6
**Confidence:** 4

**Review:**

This paper proposes two different tasks for estimating data leakage in LLMs. Their methods don't need access to the training dataset. They define two types of methods: prediction of critical keyword, and prediction of a wrong option in a multiple-choice question.
The authors report the exact match accuracy of the predicted word for the first option, and ROUGE and BLEURT score for the second option.

While the authors propose an interesting idea, there are some loose ends to the paper:
- The biggest drawback is how to interpret the scores? The LLM could be predicting these words correctly simply because they are the "most logical choices". A related question is how select a threshold.

- Another drawback is the lack of comparison and contrast with existing methods. For example, the reference [5] also proposes a similar technique. However, instead of predicting the critical keyword, it tries to predict all the text. How is the ranking induced by that score different from the ranking induced by the proposed score.

- Finally, the quality of writing should be improved. For example:
Line 29-30: To summarise what ? The sentence does not make any sense to me. Also the references in this paragraph are all towards existing LLMs.

---

### Decision · Program_Chairs · 2023-10-28

**Decision:**

Accept (Poster)

**Comment:**

An interesting research direction and a novel way of evaluating data contamination in LLM. However, clarity in writing and proper explanations for some sections need to be done well.